# One-Step Preparation of Hyperbranched Polyether Functionalized Graphene Oxide for Improved Corrosion Resistance of Epoxy Coatings

**Xuepei Miao [1,\*], An Xing [2], Lifan He [2], Yan Meng [2,\*] and Xiaoyu Li [2,\*]**

[1] School of Chemical Engineering and Materials, Changzhou Institute of Technology, No. 666 Liaohe Road, Xinbei District, Changzhou 213032, China

[2] Key Laboratory of Carbon Fiber and Functional Polymers, Ministry of Education, Beijing University of Chemical Technology, Beijing 100029, China; haxingan@hotmail.com (A.X.); helf@mai.buct.edu.cn (L.H.)

\* Correspondence: miaopei1203@126.com (X.M.); mengyan@mail.buct.edu.cn (Y.M.); lixy@mail.buct.edu.cn (X.L.)

**Abstract:** In this paper, hyperbranched polyether functionalized graphene oxide (EHBPE-GO) was prepared by a facile one-step method. Fourier transform infrared spectroscopy (FTIR), X-ray diffractometry (XRD), thermogravimetric analyzer (TGA), and trans-mission electron microscopy (TEM) results confirmed the formation of EHBPE-GO. Then, EHBPE-GO was cured with phenolic amides at room temperature to prepare epoxy coatings. The corrosion resistance of epoxy coatings was investigated systematically by using electrochemical and traditional immersion methods. Results show that a small amount of EHBPE-GO (8 wt % of Diglycidyl ether of bisphenol A (DGEBA)) in epoxy coating achieves 50% higher improvement in acid-resistance than unmodified neat DGEBA resin. For the nanocomposite epoxy coating, the superior acid-resistance is attributed to the increased crosslink density and the impermeable 2D structure of EHBPE-GO. This work provides a facile strategy to develop the effective improved corrosion resistance nanofiller for epoxy coating.

**Keywords:** hyperbranched; nanocomposites; graphene oxide; corrosion; coating

---

## 1. Introduction

Epoxy coating is widely used in various applications due to its excellent adhesion properties, low-shrinkage, good corrosion resistance, and chemical resistance [1,2]. However, their wider application is limited due to its unsatisfactory corrosion resistance. Therefore, numerous efforts have been undertaken to improve the corrosion resistance of epoxy coating. Thermoplastics [3], siloxane modifiers [4], liquid rubber [5], and hyperbranched polymers have been used to modify the epoxy coating either by chemical or physical methods. The shortcoming of thermoplastics is the increasing viscosity, which is undesirable for processing. In addition, the shortcomings of siloxane and liquid rubber are the reaction-induced phase separation, which is sensitive to processing and thus compromises the flexibility in cure scheme and processing conditions. Despite the above progress, the available modified methods can't achieve the desired corrosion resistance. Therefore, highly efficient modifiers need to urgently be developed for epoxy.

Graphene oxide (GO) has shown several excellent performances such as excellent impermeability to gases, chemical (acid/base/salt) resistance, [6] thermal properties, [7] and mechanical strength [8]. Therefore, graphene-based materials have been intensively studied in the coating industry in recent years. However, low dispersion limits its wider application. The oxygen-containing groups within its structure provide several reactive sites for the further modification of GO [9]. Hyperbranched polymers have attracted considerable attention due to their unique structure and excellent properties.

The unique structure endows them with excellent properties such as low melt, solution viscosities, and high solubility [10,11]. Therefore, when used as a filler, hyperbranched polymers show the lower viscosity of polymers, which can improve the processability and the more functional groups that can increase the crosslink density and reactivity. When combining GO and hyperbranched polymers into one, hyperbranched polymers functionalized with GO are obtained and could show many attractive properties. To this end, a great deal of research on graphene oxide functionalized by hyperbranched polymers has been reported [12–16]. More recently, Zhang [12] reported a functionalized graphene oxide with a hyperbranched cyclotriphosphazene polymer and found that the addition of modified GO can facilitate the curing reaction of dicyclopentadiene bisphenoldicyanate ester and improve the thermomechanical properties of dicyclopentadiene bisphenol dicyanate ester (DCPDCE) resin, which bears the hydrophobic cycloaliphatic backbone and possesses better dielectric properties, lower moisture absorption, superior thermal cycling tolerance, and better retention of mechanical properties at high temperature. Yan [13] reported a hyperbranched polysiloxane grafted reduced graphene oxide (HBPSi-rGO) by using the reaction of hydrosilylation, and concluded that the appropriate content of HBPSi-rGO can enhance the mechanical properties including the impact and flexural strengths of the bismaleimides (BMI) resin. Wu [14] reported a simple and effective route to hyperbranched polymer functionalized graphene sheets (GS–HBA), and found that the GS-HBA composite has higher modulus, tensile strength, and yield strength. Mahapatra [16] reported a new and facial route to prepare graphene oxide (GO) reinforced hyperbranched polyurethane (HPU) composites by the in situ polymerization technique, and found that the highly flexible graphene-based shape memory polyurethane composite exhibited higher modulus and breaking stress, and exceptional elongation-at-break. It is worth noting that most of these studies have been focused on the improvement of mechanical properties and the functionalization methods. In addition, most reported functionalization methods consist of multiple steps that are time-consuming and inefficient [14–16]. To our best knowledge, application of hyperbranched polymer functionalization of graphene oxide in a surface coating has seldom been reported in the literature [11], which may be a promising way to improve the anticorrosion performance of epoxy resin.

Herein, the present work is aimed at studying the preparation of EHBPE-GO and its potential application in surface coating. The successful functionalization was confirmed by FTIR, XRD, TGA and TEM. Then, effects of EHBPE-GO on corrosion resistance of epoxy coating were also investigated. The corrosion resistance was investigated by both electrochemical and traditional immersion methods systematically. In addition, possible anti-corrosion mechanisms are provided.

## 2. Experimental

### 2.1. Materials

Natural graphite powder (30 μm with purity >99.85%) was purchased from Sinapharm Chemical Reagent Co. Ltd., Shanghai, China. DGEBA was purchased from Yue Yang Resin Factory, China (EEW = 190.04 g/equiv.). LITE3000 was purchased from CardoliteCo. Ltd. (Jinan, China). Xylene, butanone, N,N-dimethyl formamide (DMF) and terahydrofuran (THF) were purchased from Beijing Reagent Co., China. Sodium chloride, sodium sulfate, potassium permanganate, m-dihydroxybenzene, and tetrabutylammonium bromide (TBAB) were purchased from Tianjin Fuchen Chemical Reagents Factory (Tianjin, China). 1,1,1-Trihydroxymethylpropane triglycidyl ether (TMPGE) (99%) was purchased from Titanchem Co. Ltd. (Shanghai, China). CDCl$_3$ was purchased from Beijing InnoChem Science & Technology Co. All solvents and reagents are analytical pure and were used as received. For convenience, the abbreviations of main material are tabulated in Table 1.



**Table 1.** Abbreviations of the main material.

| Full Name | Abbreviation |
| --- | --- |
| Hyperbranched polyether functionalized graphene oxide | EHBPE-GO |
| Graphene oxide | GO |
| 1,1,1-Trihydroxymethylpropane triglycidyl ether | TMPGE |
| Tetrabutylammonium bromide | TBAB |
| Diglycidyl ether of bisphenol A | DGEBA |

*2.2. Characterizations*

$^1$H NMR spectra were collected using a Bruker AV-600 spectrometer (Karlsruhe, BW, Germany) (600 MHz). CDCl$_3$ was used in NMR measurements as solvent. Number-average molecular weights (M$_n$) and the polydispersity index (PDI) of the synthesized EHBPE were determined using a Waters 515–2410 gel permeation chromatography (Milford, MA, USA) (GPC) system. Fourier transform infrared spectroscopy (FTIR) spectra were collected on a Bruker Tensor-37 spectrophotometer (Karlsruhe, BW, Germany) using the KBr disc technique. Thermal stability was measured using a PerkinElmer Pyris1 thermo gravimetric analyzer (Milford, MA, USA) (TGA) from 30 to 800 °C at a heating rate of 10 K/min under nitrogen. Transmission electron microscopy (TEM) images were taken by using a Hitachi-800 (Tokyo, Japan) with an accelerating voltage of 200 kV. The X-ray diffraction measurement was recorded at room temperature (ca. 25 °C) at scan rate 2°/min using a Rikagu X-ray diffractometer (Tokyo, Japan) (XRD, D/Max2550VB+/PC, manufactured by the Rigaku Corporation, Cu-K$\alpha$ radiation with a wavelength of $\lambda$ = 0.154056 nm) over a range of 2$\theta$ = 5–70°. Electrochemical Impedance Spectroscopy (EIS) is a very effective method for evaluating the corrosion resistance of organic coating. [17–21] EIS was conducted on a Zahner electrochemical work station (Kronach, BY, Germany,) using a three-electrode system with a frequency range of 100 KHz to 0.1 Hz. The coated panel has an exposed area of approximately 10 cm$^2$ served as the working electrode, saturated calomel electrode (SCE) as reference electrode and a platinum as auxiliary electrode. Electrolyte solution was a 3.5% NaCl solution. The amplitude of the signal was 10 mV. Electrochemical impedance measurements were carried under open circuit potential at room temperature (~25 °C). The results of EIS analysis for the epoxy coatings are presented as Bode plots. Bode plots are comprised of a graph between logmagnitude versus frequency, measured over a range of frequencies. The high frequency region of the Bode plot provides information about coating defects and other changes in surface area; the low frequency region provides information about processes occurring near the metal surfaces. Potentiodynamic polarization curves were obtained by changing the electrode potential in the range of 500 mV around the open-circuit potential (OCP) against SCE at a scan rate of 1.0 mV·S$^{-1}$.

*2.3. Preparation of GO*

Graphene oxide was produced from natural graphite powder by a modified Hummer's method [22–24] and ultrasonication. In a typical process, an amount of 2 g graphite powder and 1 g sodium chloride were added into three necked round bottom flasks in an ice-bath. Subsequently, 40 mL high concentrated acid including concentrated sulphuric acid (98%) and phosphoric acid with a volume ratio of 9:1 were gradually added into the three necked round bottom flasks under a magnetic stirrer. While maintaining vigorous agitation, in order to prevent the temperature of the suspension from exceeding 20 °C, potassium permanganate (6 g) was carefully added into the suspension. The ice-bath was then removed and the mixture was stirred at 40 °C for 12 h in a water bath. After cooling to room temperature, the mixture was gradually added into a 280 mL ice water containing 2 mL 30% H$_2$O$_2$ solution. The resulting GO suspension was washed by repeated centrifugation, first with 5% of HCl solution to remove the excess manganese salt followed by water until the pH of the solution became neutral. Finally, stable water-dispersion of such purified GO was prepared ultrasonically and stored for utilization.

## 2.4. Preparation of EHBPE

TMPGE (36.28 g, 0.12 mol), m-dihydroxybenzene (4.40 g, 0.04 mol) and TBAB (1.93 g, 0.006 mmol), which is fixed to 5% of the moles of TMPGE, were mixed in a flask and stirred at 100 °C for 48 h under $N_2$ protection. After cooling to room temperature, 100 mL THF was added into the flask and then precipitated into hot water. The precipitate was then dissolved in ethanol, dried with anhydrous $Na_2SO_4$ overnight, and filtered; the filtrate was stirred into ether, followed by removing the THF in vacuum to obtain a viscous liquid. The obtained products were characterized using NMR and FTIR. $^1H$ NMR (CDCl$_3$, d): δ 7.15, 6.50 (CH, aromatic), 4.12–3.36 (br, CH and CH$_2$) 3.11 (CH, epoxide), 2.77 (CH$_2$, epoxide), 2.57 (CH$_2$, epoxide), 1.39–1.23 (br, CH$_2$), 0.85–1.10 (br, CH$_3$, trisubstituted B$_3$). FTIR υ (cm$^{-1}$): 3440 (br, OH); 2910–2880, 1660, 1600, 1470, 1280, 1110, 906 (CH$_2$, epoxide);841, 760 cm$^{-1}$ (CH, epoxide). T$_g$: −23.12 °C [25].

## 2.5. Preparation of EHBPE-GO

It is known that GO possesses various excellent properties; however, the aggregation of GO limits its wider application. In order to improve its dispersity in solvent, GO was functionalized with EHBPE. In this paper, EHBPE-GO was prepared by a simple one-pot polymerization technique. The synthetic strategy of EHBPE-GO is shown in Scheme 1. Detailed information concerning the synthesis are shown below. In a typical experiment, graphene oxide (10.2 mg) and EHBPE (1.019 g) were first dispersed in DMF (40 mL) by an ultrasonicator at room temperature for 30 min, then TBAB (0.05 g) was added into the mixture. Subsequently, the mixture was heated to 80 °C under magnetic stirring. After reaction for 10 h, the desired EHBPE-GO was obtained by washing DMF and catalyst with deionized water.

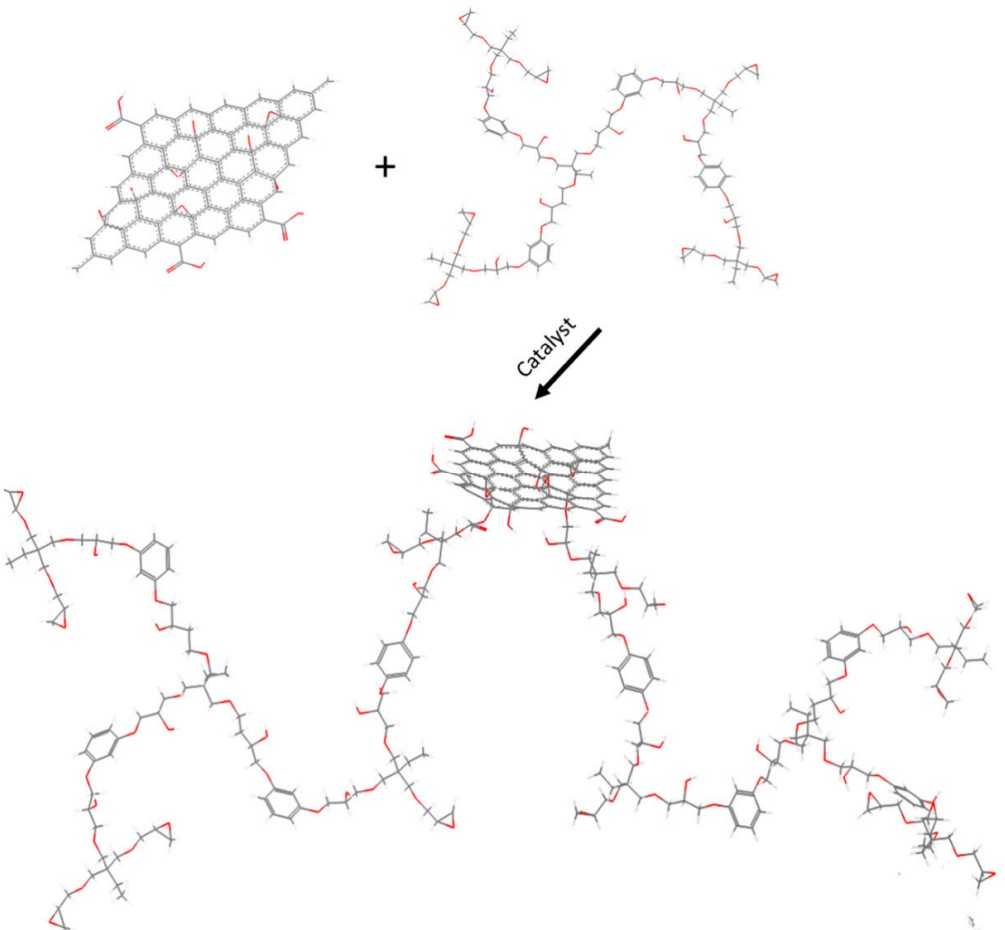

**Scheme 1.** Synthesis of EHBPE-GO.

*2.6. Preparation of EHBPE-GO/DGEBA Nanocomposite Epoxy Coating*

The metal matrix was Q235 mild carbon steel panels (100 mm × 50 mm × 1 mm). The steel panels were polished using 400# abrasive paper to remove mill scale, then were cleaned with acetone and ethanol successively. The main compositions of the coatings are as follows: 33.4% hybrid epoxy (EHBPE-GO: DGEBA = (0%, 3%, 5%, 8%, 10%) (m:m)), 6% zinc molybdate (800 mesh), 20% iron oxide red (800 mesh), 8% zinc phosphate (800 mesh), 4.8% mica power (1250 mesh) and 7.8% talcum powder (800 mesh) and 20% mixed solvents of xylene and butanone with a ratio of 7:3 (v:v). The main process for preparing epoxy coating consists of three steps: (1) Create a grind paste. Firstly, resins and solvents were mixed under magnetic stirring until the mixture became uniform and transparent. Then, pigments were added into the mixture under mechanical stirring. Subsequently, the obtained mixture was added into a cone mill to create a grind paste in which pigments are uniformly dispersed. (2) Formulate the paint. Cure agent (LITE3000) (CardoliteCo. Ltd., Jinan, China,), mixed solvent, and the above obtained grind paste were added into a beaker under mechanical stirring until the mixture became uniform and transparent. (3) Curing process. The paints were coated on the steel substrate on both sides, samples were cured at room temperature for a week. The thickness is about 30 μm. For comparison, neat epoxy specimen was fabricated in the same manner.

*2.7. Corrosion Testing*

In order to accelerate the failure process of the coatings, 10% $H_2SO_4$ solution, 5% NaOH solution, and deionized water were chosen to perform the corrosion test. The coated samples were immersed in 10% $H_2SO_4$ solution, 5% NaOH solution, and deionized water, 7 cm depth to the solution surface. The maximum immersion testing time was up to 80 days.

## 3. Result and Discussion

*3.1. Characterization of EHBPE-GO*

3.1.1. FTIR Characterization

In order to make sure the successful functionalization of EHBPE, FTIR was employed to investigate the linkage of EHBPE and GO. As shown in Figure 1, the absorbance peak at ~1750 cm$^{-1}$ represents the carbonyl moieties on the surface of GO, which is in good agreement with the reported studies [26]. Peaks at 2960–2880 cm$^{-1}$ belong to the aliphatic –C–H stretching vibration, peaks at about 1231 and 1100 cm$^{-1}$ can be assigned to Ph–O–C and C–O–C stretching vibration, respectively, and the characteristic absorption of epoxy groups can be observed at about 908 and 843 cm$^{-1}$, indicating the successful functionalization of GO. Moreover, shifting of the peaks at 3441$^{-1}$ (–OH stretching vibration) to a lower wavenumber is direct evidence of H-bonding generation between the EHBPE and GO [27].

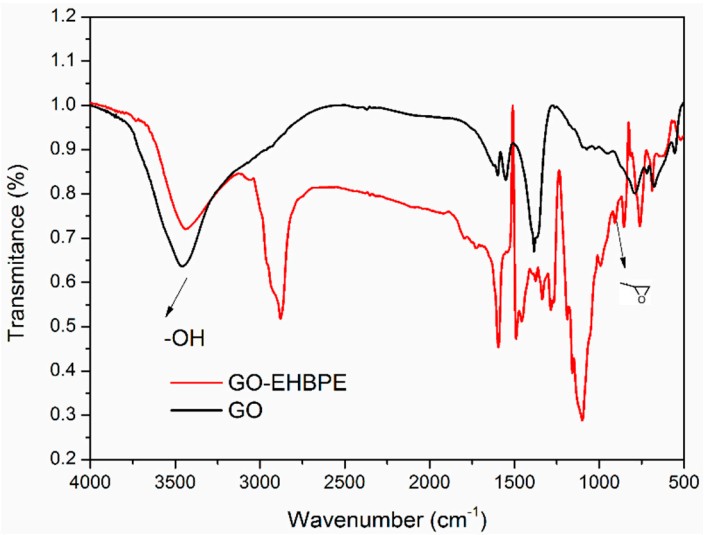

**Figure 1.** FTIR spectra of GO and EHBPE-GO.

### 3.1.2. TGA Characterization

TGA could give lots of information about chemical composition, structure, and thermal stability of materials. Therefore, GO and EHBPE-GO were analyzed by TGA, with the results shown in Figure 2. As for GO, weight loss around 100 °C is closely related to the evaporation of stored water from its layer structure [28]. Additionally, an obvious weight loss can be found in the temperature range of 100 to 250 °C, which can be explained by the decomposition of oxygen-containing functional groups including hydroxyl, epoxy groups, and carboxyl. For EHBPE-GO, a major weight loss above 320 °C can be found, which indicates an enhancing thermal stability of EHBPE-GO, compared with GO. Compared with EHBPE, EHBPE-GO decomposed at a higher temperature, and it can also be found that EHBPE-GO has higher residue yield at 600 °C, which further provides evidence for the successful preparation of EHBPE-GO.

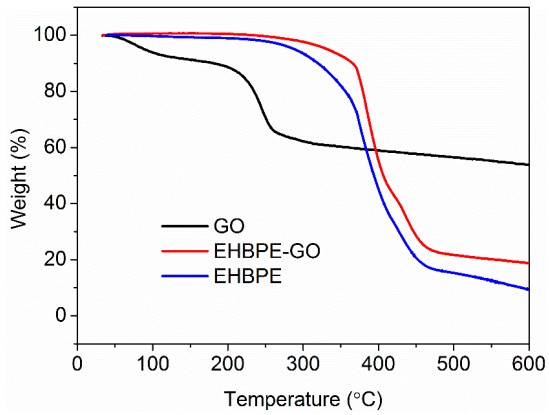

**Figure 2.** TGA curves of GO, EHBPE, and EHBPE-GO.

### 3.1.3. XRD Characterization

XRD diffraction patterns of GO and EHBPE-GO are shown in Figure 3. As shown in Figure 3, the value at about $2\theta = 10.18°$, corresponding to an interlayer spacing of 0.87 nm (calculated using Bragg's law) owing to the presence of oxygen containing functional groups, which is a typical value for GO. A broad diffraction peak at $2\theta = 20.21°$ was observed for EHBPE-GO. This may be due to the strong interaction of well dispersed GO with polar groups of EHBPE that reduces its crystallinity to a large extent. [29,30], which conformed to the structural properties of a hyperbranched polymer [31].

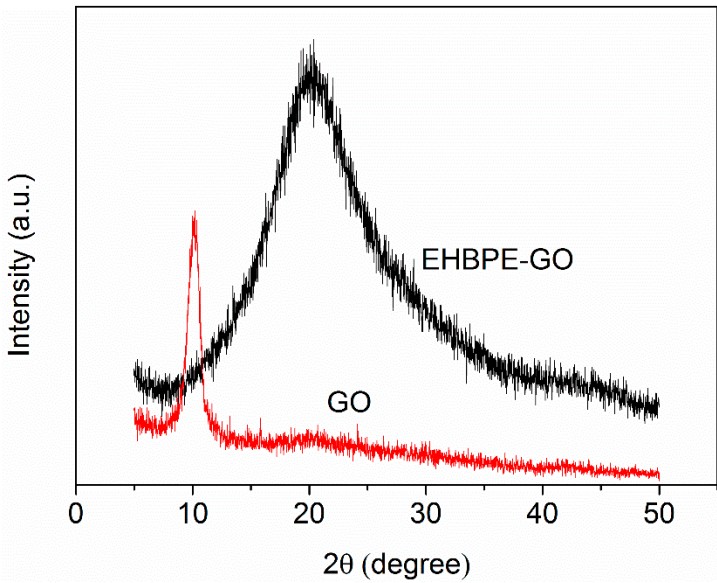

**Figure 3.** XRD patterns of GO and EHBPE-GO.

### 3.1.4. TEM Characterization

In order to further confirm the successful functionalization of GO, the nanosheets morphology of GO and EHBPE-GO were investigated by the TEM technique, with the results shown in Figure 4. TEM analysis (Figure 4a) shows that the GO nanosheets are very thin and have some wrinkles, but the surface of the GO nanosheets is fairly smooth, which is consistent with the morphology typically reported in the literature [14,15,32–34]. In contrast, EHBPE-GO (Figure 4b) exhibits a different morphology. Compared with GO, the nanosheets of EHBPE-GO appear to be rough, and clearly some black regions covered on the surface of EHBPE-GO can be observed. This can be attributed to the grafting of hyperbranched polyether.

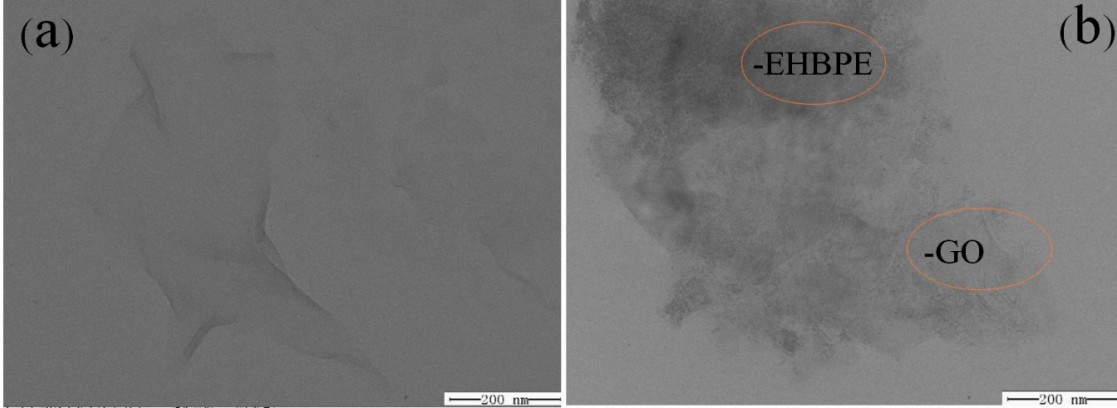

**Figure 4.** TEM images of (**a**) GO and (**b**) EHBPE-GO.

### 3.2. EIS Characterization

An effective way to evaluate the corrosion resistance of organic coating is EIS characteration. It is known that the impedance value at low frequencies in the Bode plot is always used to evaluate the corrosion resistance of the coatings [35,36]. The higher the value of impedance is, the higher the corrosion resistance. Effects of EHBPE-GO loading on the corrosion resistance of hybrid coating were investigated, and the results are shown in Figure 5. It can be found that nanocomposite epoxy coating shows superior corrosion resistance to that of neat DGEBA coating, indicating that EHBPE-GO can significantly improve the corrosion resistance of epoxy coating. In addition, when EHBPE-GO loading

increases, hybrid coating impedances (in low frequency domain) first increase then decrease; and the best overall performance is achieved at 8% loading. Therefore, for simplicity, when comparing effects of exposure time on the corrosion resistance of hybrid coating, the loading amount is fixed to 8% EHBPE-GO loading.

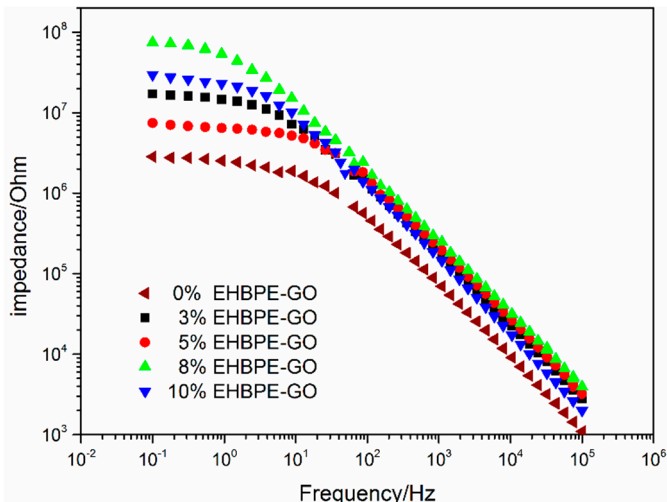

**Figure 5.** Electrochemical Impedance Spectroscopy (EIS) spectra Bode plots of hybrid coatings at different EHBPE loadings after immersion in 3.5% NaCl solution for five days.

EIS results obtained in short exposure time are not enough to reveal the protective performance of the nanocomposite epoxy coating. Thus, to investigate whether the coatings can also serve as an anticorrosion coating in 3.5 wt % NaCl solution over a much longer time scale, the maximum immersion testing time was up to 80 days. EIS data of epoxy coatings after different exposure time in corrosion environment are shown in Figure 6. In Figure 6a, it is obvious that impedances at 0.1 Hz are larger than $10^7$ $\Omega$ cm$^2$ after 80 days of immersion, indicating that such nanocomposite epoxy coatings can shield metals well. Interestingly, during the immersion testing, impedance values at low frequencies increase rather than decrease, which is inconsistent with the traditional phenomenon. This can be explained by the formation of the corrosion product films which can act as a protective layer due to the reaction between corrosive ions and metals. No matter how compact the coating is, some corrosion molecules or ions ($H_2O$, $O_2$ and $Cl^-$) can penetrate through the matrix over a long-time scale [37]. Once the steel substrate is corroded, corrosion molecules or ions can react with the steel substrate. By now, the corrosion product will form a protective layer. Correspondingly, Bode phase plots show the same trend as Bode modulus plots. Effects of exposure time on the corrosion depth of epoxy coatings were investigated, and the results are shown in Figure 7. For DGEBA coating, corrosion depth increases with immersion time. It is obvious that the corrosion depth of DGEBA coating was higher than that of nanocomposite epoxy coating containing 8% EHBPE-GO. For nanocomposite epoxy coating containing 8% EHBPE-GO, as the immersion time going on, corrosion depth is almost constant, indicating the excellent corrosion resistance during the 80 days of immersion.

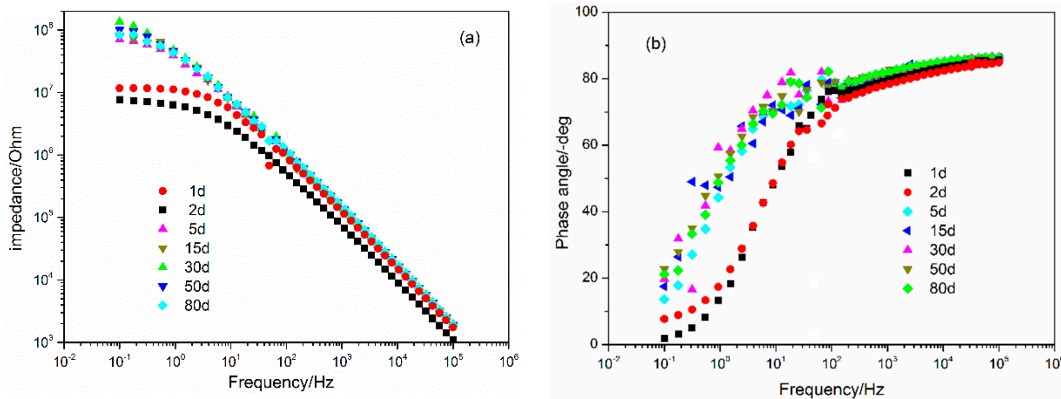

**Figure 6.** (**a**) Bode modulus plots, and (**b**) Bode phase plots of the hybrid coating containing 8% EHBPE-GO after immersion for a different number of days.

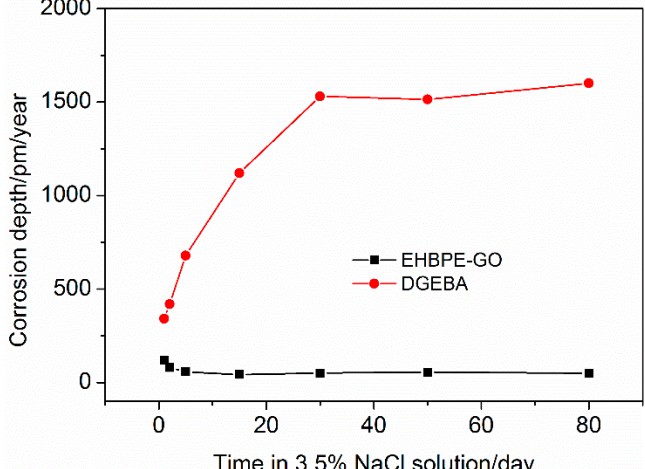

**Figure 7.** Corrosion depth vs. immersion time in 3.5% NaCl solution.

## 3.3. Potentiodynamic Polarization Test

In addition, the corrosion resistance of epoxy coatings was also evaluated through the potentiodynamic polarization technique. Corrosion potential, corrosion current, and the polarization resistance after immersion in 3.5% NaCl solution for 80 days were calculated from the measured potentiodynamic polarization curves and summarized in Table 2. Figure 8a shows the measured potentiodynamic polarization curves in a 3.5% NaCl solution. During the immersion time, the potential of nanocomposite epoxy coating moves towards the positive potential, which is in agreement with EIS results. Similar to the arguments used to explain the increased impedance at low frequencies with the increasing immersion time, the potential of nanocomposite epoxy coating moves towards the positive potential can also be attributed to the formation of the corrosion product films. From Figure 8b, we can find that the potential of nanocomposite epoxy coating moves towards the positive potential, compared with DGEBA epoxy coating, and DGEBA epoxy coating has higher current density than nanocomposite epoxy coating, suggesting that the nanocomposite epoxy coating has a better corrosion resistance than the neat DGEBA coating, which is also consistent with EIS results.

**Table 2.** Electrochemical performance parameters of epoxy coatings after immersed in 3.5% NaCl solution for 80 days.

| Sample Code | Ec (V) | Ic (nA·cm$^2$) | R$_p$ (MΩ·cm$^2$) |
|---|---|---|---|
| DGEBA | −0.23 | 10.90 | 5.13 |
| EHBPE-GO/DGEBA | −0.18 | 0.14 | 0.64 |

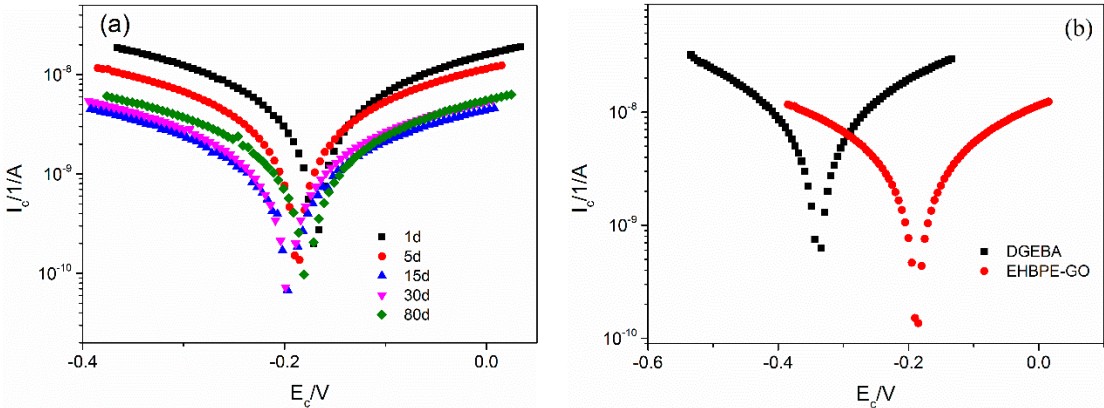

**Figure 8.** (**a**) polarization curves for nanocomposite epoxy coating after different immersion time; (**b**) polarization curves for nanocomposite epoxy coating and DGEBA coating after 30 days of immersion.

*3.4. Corrosion Testing*

A traditional immersion test offers a simple and cheap way to investigate the long-term scale corrosion resistance of nanocomposite epoxy coating. In this study, an immersion test was carried out on three corrosive solutions, i.e., 10% H$_2$SO$_4$ solution, 5% NaOH solution, and deionized water. In addition, a salt spray test was also investigated and results are summarized in Table 3. From Table 3, we can find that both DGEBA epoxy coating and nanocomposite epoxy coating show superior resistance to deionized water and 5% NaOH solution. However, significant differences exist in resistance to 10% H$_2$SO$_4$ solution. Compared with DGEBA epoxy resin coating, the acid-resistance of the nanocomposite epoxy coating is improved markedly up to 50%.

**Table 3.** Corrosion test results of epoxy coatings.

| Sample Code | 5% NaOH (day) | 10% H$_2$SO$_4$ (day) | H$_2$O (day) | Salt Spray Test (day) |
|---|---|---|---|---|
| DGEBA | 80 | 8 | 80 | 80 |
| EHBPE-GO/DGEBA | 80 | 12 | 80 | 70 |

It is known that some micro-pores always exist even in very highly crosslinked networks. When the coating film was immersed in the corrosive solution, paths formed by relatively big micro-pores provide pathways for corrosive substance to diffuse through (see Figure 9 below). It is believed that the wider the diffusion path, the weaker the corrosion resistance of the material. It has been reported that when hyperbranched polymers are used as modifiers in composites, crosslink density can be significantly enhanced due to its multi-functional groups. Hyperbranched polymers have several end groups, which can increase the crosslink points of epoxy network. Thus, when hyperbranched polymers participate in the formation of epoxy network, the crosslink density will be increased. Correspondingly, the diffusion path will be narrowed down. Therefore, it can be concluded that the diffusion path of neat DGEBA coating is wider than that of nanocomposite epoxy coating containing EHBPE-GO due to the higher crosslinking density in the nanocomposite epoxy coating. Once the coating was penetrated by an electrolyte solution, the corrosion process is dramatically accelerated, which increases the corrosion rate. In summary, the lower corrosion rate of nanocomposite epoxy coating is mainly due

to its narrower diffusion path, which can shield corrosive substances. In addition, as discussed in the Introduction, GO has various excellent properties due to its uniquely impermeable 2D structure that can exhibit an exceptional barrier to corrosion ions or molecules, thus providing a physical separation between the steel substrate and electrolyte solution (as shown in Figure 9), leading to an excellent corrosion resistance of nanocomposite epoxy coating. It is worth noting that the beneficial role of curing agent LITE3000 compatible with all components of epoxy compositions/varnishes with GO may be another key factor [38]. To sum up, the excellent corrosion resistance of nanocomposite epoxy coating obtained can be explained by the more compact network structure due to the good impermeability of GO and the high crosslink density of EHBPE, and the compatibility of all components.

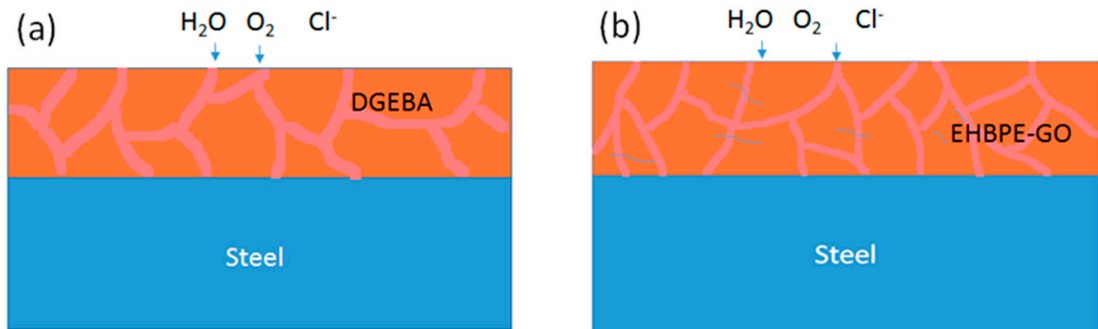

**Figure 9.** Anti-corrosion process of (**a**) DGEBA and (**b**) nanocomposite epoxy coating.

## 4. Conclusions

In summary, we have presented a facile approach to prepare EHBPE-GO. The successful preparation of EHBPE-GO was investigated by FTIR, XRD, TGA, and TEM. Results show that EHBPE-GO can improve corrosion resistance significantly when used as an epoxy coating modifier. Compared with DGEBA epoxy resin coating, the acid-resistance of the nanocomposite epoxy coating is improved markedly up to 50%. The improved corrosion resistance can be explained by the more compact network structure due to the good impermeability of GO and the high crosslink density of EHBPE.

**Author Contributions:** X.M. and A.X. conceived and designed the experiments; X.M. performed the experiments; X.L. and Y.M. analyzed the data; L.H. contributed reagents/materials/analysis tools; X.M. wrote the paper.

**Funding:** This research was funded by the Natural Science foundation of the Jiangsu Higher Education Institutions of China, grant number 19KJB430009 and the National Science Foundation of China, grant number 51173012.

**Conflicts of Interest:** The authors declare no conflict of interest.

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
