# Peer review of "One-Step Preparation of Hyperbranched Polyether Functionalized Graphene Oxide for Improved Corrosion Resistance of Epoxy Coatings"

_coatings, doi:10.3390/coatings9120844_

Round 1

Reviewer 1 Report

Manuscript on epoxy coatings with hyperbranched-polymer-functionalized-GO. Their anticorrosive properties were confirmed by proper methods. In my opinion, this article deserves to be published in Coatings. However, please, take into account the beneficial role of curing agent LITE3000 in compatibility of all components of epoxy compositions/varnishes with GO. Such cardanol-based amines had been proved to be chemically compatible with carbon nanofillers: graphene and carbon nanotubes [S. Kugler, K. Kowalczyk, T. Spychaj, Influence of synthetic and bio-based amine curing agents on properties of solventless epoxy varnishes and coatings with carbon nanofillers, Prog. Org. Coat. 109 (2017) 83–91. doi:10.1016/j.porgcoat.2017.04.033.]. Please, include this fact in Results&discussion section.

Other comments:

-please, correct the format of Y values in Fig. 8 to scientific format, like in Figs 5 and 6

-please, check minor language mistakes, like in line 229: results is not (it should be: results are not), line 15: thermogravimetric

Author Response

Responses to reviewers’ comments

Dear editor and reviewer:

Thank you for the comments. We have revised our manuscript accordingly, and the revised and added parts are shown in red. Every comment has been seriously considered and reflected in the revised manuscript. Detailed responses to each comment are described below.

Comment. 1: Following points need to be taken care of (some points are highlighted with yellow)–

Despite the above progress, the available modifies cannot achieve- correct it.

Response: As suggested. the sentence has been corrected. The revised sentence is as follows: “Despite the above progress, the available modified methods cannot achieve the desired corrosion resistance.”

Is it epoxide-terminated (EHBPE)?

Response: Yes, it is epoxide-terminated (EHBPE).

“ Epoxy coating is widely used in various applications due to its excellent adhesion properties, low-shrinkage, good corrosion resistance and chemical resistance [1,2]. However, their wider 27 application is limited due to its unsatisfactory corrosion resistance.” – epoxy is known for good corrosion resistance property. Can you be more clear on the unsatisfactory corrosion resistance of epoxy?

Response: Thank you for pointing that out. What we meant is when high corrosion resistance is required, typical epoxy formulation is not completely satisfactory

Define properties of DCPDCE resin.

Response: As suggested, the properties of DCPDCE resin have been difined, and the Corresponding changes are made as follows: Zhang [12] reported a functionalized graphene oxide with a hyperbranched cyclotriphosphazene polymer, and found that the addition of modified GO can facilitate the curing reaction of dicyclopentadiene bisphenoldicyanate ester and improve the thermomechanical properties of DCPDCE resin which bearing the hydrophobic cycloaliphatic backbone and possess better dielectric properties, lower moisture absorption, superior thermal cycling tolerance, and better retention of mechanical properties at high temperature.

“ To our best knowledge, application of hyperbranched 62 polymer functionalization of graphene oxide in surface coating have been seldom reported in the 63 literature, which….” Authors should cite appropriate literature.

Response: Thank you for pointing that out. As far as we know, there were no reports on the application of hyperbranched  polymer functionalization of graphene oxide in surface coating.

L 65: “..preparation of epoxide EHBPE-GO and..” Check it

Response: Thank you for pointing that out. The sentence of  “..preparation of epoxide EHBPE-GO and..”  has been revised as “…preparation of EHBPE-GO and..” in the revised version.

L 85: “……EHBPES were..” Check it

Response: Thank you for pointing that out. “S” has been deleted in the revised version.

8.Traditional immersion test offers a simple and cheap methods to..”

Response: Thank you for pointing that out. “a simple and cheap methods” has been revised to  “a simple and cheap way”.

ref#12- title included?

Response: Thank you for pointing that out. The title of ref#12 has been deleted.

L-19: “…small amount of EHBPE-GO (8 wt% of DGEBA) in epoxy coating achieves 50% 19 high improvement in acid-resistance.”- What was the reference material used in this comparative study?

Response: The reference material used in this comparative study was unmodified neat DGEBA resin.

Comment. 2: 1. Can the authors elaborate the advantages of hyperbranched polymers as filler?

Response: The advantages of hyperbranched polymers as filler are as follows: the lower viscosity of polymers which can improve the processability; the more functional groups that can increase the crosslink density and reactivity.

2.The unique structure endows them with excellent properties such as low 43 melt, solution viscosities, and high solubility ..”- Elaborate solubility and low viscosities property

Response: What we mean is when comparing with their linear analogues, hyperbranched polymers possess considerably lower viscosity, better solubility.

3.“ When combining GO and hyperbranched polymers into one, hyperbranched polymers functionalized with GO are obtained and will show 45 many attractive properties…”- what are the attractive properties?

Response: The attractive properties such as excellent dispersibility due to the functionalization of hyperbranched polymer, and barrier property brought by GO.

 Comment. 3

1: Authors need to indicate peaks of each functional groups in the spectra

Response: As requested, peaks of each functional groups have been indicated in Fig.1. In addition, the detailed attribution of each peak is given in section 2.4.

Fig. 1 FTIR spectra of GO and EHBPE-GO.

“GO. Moreover, shifting of the peaks at 3441-1 (-OH stretching vibration) 175 to lower wavenumber is a direct evidence of H-bonding generation between the EHBPE and..” – Cite relevant literature to support this statement

Response: As requested, relevant literature has been added in the revised version.  The added reference is as follows: “T. Zhou, F. Li, Y. Fan, W. Song, X. Mu, H. Zhang, and Y. Wang. Chem. Commun., 2009, 3199-3201.”

Comment 4. Fig. 3- Include the XRD pattern of pristine EHBPE. Authors can include all the XRD patterns in a single graph for a comparative view.

Response:  As requested, Fig. 3 has been redrew to make a comparative view. The revised Fig. 3 is as follows:

Fig. 3 XRD patterns of GO and EHBPE-GO.

Comment 5. Figure 4. TEM images of (a) GO and (b) EHBPE-GO. – TEM images are not of good quality. Include a better resolution of TEM images in the manuscript. Indicate the GO and the epoxy matrix in the fig.

Response: Thank you for pointing that out. We feel very sorry for not providing the higher quality image due to the limited time. We believe that information about GO and EHBPE-GO can be obtained from those original images, although the TEM images are not of good quality. In addition, the GO and the epoxy matrix in Figure 4 has been indicated. The revised Fig. 4 is as follows:

Fig. 4 TEM images of (a) GO and (b) EHBPE-GO.

Comment 6. L 219: “It can be found that nanocomposite epoxy 219 coating shows superior corrosion resistance than that of neat…”- How much is the superior performance? The current schematic Fig. 9 does not show any clear message.

Response: It is known that the higher the impedances at low frequency, the better the corrosion resistance of the coating material. Therefore, as shown in Fig. 5, the impedance of nanocomposite epoxy coating is one order larger than that neat DGEBA coating. The detailed description of Fig.9 will be given in response to Comment 8.

Comment 7: Coatings industry follows ASTM standard corrosion testing for evaluating coatings performance against corrosion. Is this traditional immersion test accepted by the coatings Industry?

Response: As far as we know, the traditional immersion test has been accepted by most of the coating enterprises, especially for the enterprises that supplies for heavy anticorrosive field.

Comment 8: Figure 9. Anti-corrosion process of (a) DGEBA and (b) nanocomposite epoxy coating”- Not meaningful. Can the authors explain the mechanism and the schematic in depth? Authors are recommended to elaborate the anti-corrosion mechanisms.

Response: As requested, anti-corrosion mechanisms has been elaborated, and the added parts are as follows: “It is known that some micro-pores always exist even in very highly crosslinked network. When the coating film was immersed in the corrosive solution, paths formed by relatively big micro-pores provide pathways for corrosive substance to diffuse through (see Fig. 9 below). It is believed that the wider the diffusion path, the weaker the corrosion resistance of the material. It has been reported that when hyperbranched polymers are used as modifiers in composites, crosslink density can be significantly enhanced due to its multi-functional groups. Hyperbranched polymers have several end groups, which can increase the crosslink points of epoxy network. Thus, when hyperbranched polymers participate in the formation of epoxy network, the crosslink density will be increased. Correspondingly, the diffusion path will be narrowed down. Therefore, it can be concluded that the diffusion path of neat DGEBA coating is wider than that of nanocomposite epoxy coating containing EHBPE-GO due to the higher crosslinking density in the nanocomposite epoxy coating. Once the coating was penetrated by electrolyte solution, the corrosion process is dramatically accelerated, which increases the corrosion rate. In summary, the lower corrosion rate of nanocomposite epoxy coating is mainly due to its narrower diffusion path which can shield corrosive substances. ”

Fig. 9 Anti-corrosion process of (a) DGEBA and (b) nanocomposite epoxy coating.

Comment 9 “..It is known that GO possesses various excellent properties, however, the aggregation of GO limits its wider application…”- What are the main reasons for GO aggregation?

Response: The main reasons for GO aggregation is because of the π–π stacking and hydrophobic interactions.

Is there a limit on the %Wt. of GO to be used in this epoxy coatings? How was the dispersion quality of nanosheets in epoxy coating? Highlight it.

Response:  Effects of EHBPE-GO on the corrosion resistance of epoxy coating were investigated. And the results show that the best overall performance is achieved at 8% loading. However, when the EHBPE-GO loading increases 10%, the corrosion resistance of epoxy coating decrease. Therefore, there is a limit on the content of GO to be used in this epoxy coatings.

Comment 10: Compared with DGEBA epoxy resin coating, the acid-resistance 277 of the nanocomposite epoxy coating are improved markedly up to 50%. Why does the nanocomposite epoxy performed better in acid resistance test but showed comparable results in deionized water and NaOH solution?

Response:  That is a good point. Epoxy coating shows excellent alkali-resistance and water resistance due to the ether bonds in the molecular structure. However, the acid resistance of epoxy coating is not so good as alkali-resistance and water resistance. The addition of EHBPE-GO contributes to the acid resistance of nanocomposite epoxy coating during the investigated time scale.  If the investigated time increase, the beneficial effects of hyperbranched polymers may be more obvious.

Comment 11: L 301- “..O and the high crosslink density of 301 EHBPE.”- How did the authors came to conclude the highlighted conclusion? Can the authors provide any evidence of increased crosslinking density in the nano composite of EHBPE?

Response: In our previous study, the crosslink density of hybrid coatings was investigated by low field NMR, and the results indicating that the addition of EHBPE contributes to the increase of crosslink density. Therefore, we make the conclusion that the improved corrosion resistance can be explained by the more compact network structure due to the good impermeability of GO and the high crosslink density of EHBPE. However, it is not easy to investigate the crosslink density in the system that discussed in this paper due to the complexity of components. Therefore, there are no direct evidence for the increased crosslinking density in the nano composite of EHBPE. That is one of direction that we will continue to investigate.

Reviewer 2 Report

Please find the attachment. Thank you.

Author Response

Responses to reviewers’ comments

Dear editor and reviewer:

Thank you for the comments. We have revised our manuscript accordingly, and the revised and added parts are shown in red. Every comment has been seriously considered and reflected in the revised manuscript. Detailed responses to each comment are described below.

Manuscript on epoxy coatings with hyperbranched-polymer-functionalized-GO. Their anticorrosive properties were confirmed by proper methods. In my opinion, this article deserves to be published in Coatings. However, please, take into account the beneficial role of curing agent LITE3000 in compatibility of all components of epoxy compositions/varnishes with GO. Such cardanol-based amines had been proved to be chemically compatible with carbon nanofillers: graphene and carbon nanotubes [S. Kugler, K. Kowalczyk, T. Spychaj, Influence of synthetic and bio-based amine curing agents on properties of solventless epoxy varnishes and coatings with carbon nanofillers, Prog. Org. Coat. 109 (2017) 83–91. doi:10.1016/j.porgcoat.2017.04.033.]. Please, include this fact in Results&discussion section.

Response: Thank you for pointing that out, and we have added related references in the revised manuscripts, which are shown in red in the revised version in page 15.  The added parts are as follows: “It is worth of noting that the beneficial role of curing agent LITE3000 in compatibility of all components of epoxy compositions/varnishes with GO may be another key factor [38].”

Other comments:

1.please, correct the format of Y values in Fig. 8 to scientific format, like in Figs 5 and 6

Response: Thank you for pointing that out, Fig. 8 has been redrew and shown as follows:

Fig. 8 (a) Polarization curves for nanocomposite epoxy coating after different immersion time; (b) polarization curves for nanocomposite epoxy coating and DGEBA coating after 30 days of immersion.

please, check minor language mistakes, like in line 229: results is not (it should be: results are not), line 15: thermogravimetric

Response:  Thank you for pointing that out, “results is not” has been replaced by “results are not”, and “thermo gravimetric” has been replaced by “thermogravimetric”. The language mistakes have been checked and corrected carefully.

Reviewer 3 Report

Memorandum

Subject: Review, November 15, 2019

Journal of Coatings

Title: One-Step preparation of hyperbranched polyether functionalized graphene oxide for improved corrosion resistance of epoxy coatings

   Xuepei Miao,*a An Xing,b LifanHe,b Yan Meng,*b and Xiaoyu Li*b

a Changzhou Institute of Technology, school of chemical engineering and materials, No. 666 Liaohe Road,

   Xinbei District, Changzhou, Jiangsu 213032, P. R. China. E-mail address: miaopei1203@126.com (Xuepei

Miao)

b Key Laboratory of Carbon Fiber and Functional Polymers, Ministry of Education, Beijing University of

   Chemical Technology, Beijing 100029, P. R. China.

* Correspondence: Correspondence:-mail address: mengyan@mail.buct.edu.cn (Yan Meng);

   lixy@mail.buct.edu.cn (Xiaoyu Li)

Comments:

The authors should consider adding a nomenclature to identify the parameters and abbreviations used throughout the paper. Section “Preparation of EHBPE”, the authors have listed several constituents; as listed, they lead to some confusion. They will be more beneficial to the reader if they were tabulated. Figure 4 needs to be labeled such as the key point of interest are noted on the image.

Overall the paper is in god The paper needs some work, addressing the above minor issues should make it ready for publication.

Author Response

Comments from reviewer #3

The authors should consider adding a nomenclature to identify the parameters and abbreviations used throughout the paper. Section “Preparation of EHBPE”, the authors have listed several constituents; as listed, they lead to some confusion. They will be more beneficial to the reader if they were tabulated. Figure 4 needs to be labeled such as the key point of interest are noted on the image.

Overall the paper is in god The paper needs some work, addressing the above minor issues should make it ready for publication.

Dear editor and reviewer:

Thank you for the comments. We have revised our manuscript accordingly, and the revised and added parts are shown in red. Every comment has been seriously considered and reflected in the revised manuscript. Detailed responses to each comment are described below.

Response:  Thank you for pointing that out, the abbreviation of main material are tabulated in Table 1.and the added parts are as follows: “For convenience, the abbreviation of main material are tabulated in Table 1. “

Table 1 Abbreviation of main material.

Full name

Abbreviation

Hyperbranched polyether functionalized graphene oxide

EHBPE-GO

Graphene oxide

GO

1,1,1-Trihydroxymethylpropane triglycidyl ether

TMPGE

Tetrabutylammonium bromide

TBAB

Diglycidyl ether of bisphenol A

DGEBA

Figure 4 needs to be labeled such as the key point of interest are noted on the image

Response: Thank you for pointing that out, Figure 4 has been redrew and shown as follows:

Fig. 4 TEM images of (a) GO and (b) EHBPE-GO.

Round 2

Author Response

Responses to reviewers’ comments Dear editor and reviewer: Thank you for the comments. We have revised our manuscript accordingly, and the revised and added parts are shown in red. Detailed responses to each comment are described below. Comment 5. “ To our best knowledge, application of hyperbranched 62 polymer functionalization of graphene oxide in surface coating have been seldom reported in the 63 literature, which….” Authors should cite appropriate literature. Response: Thank you for pointing that out. As far as we know, there were no reports on the application of hyperbranched polymer functionalization of graphene oxide in surface coating. Authors stated " To our best knowledge, tr application of hyperbranched polymer functionalization of graphene oxide in surface coating have been seldom reported in the literature ,.." Hence, authors should cite appropriate literature. Response: As suggested. appropriate literature has been added in the revised version. Corresponding changes are made on page 3, line 14. [11 X Miao, A Xing, W Yang, L He, Y Meng and X Li. React. Funct. Polym., 2018, 122,116-122.] Comment 5. Figure 4. TEM images of (a) GO and (b) EHBPE-GO. – TEM images are not of good quality. Include a better resolution of TEM images in the manuscript. - Thank you for pointing that out. We feel very sorry for not providing the high-quality image due to the limited time (The revised file will be uploaded within 10 days). We believe that information about GO and EHBPE-GO can be obtained from those original images, although the TEM images are not of good quality. In addition, the GO and the epoxy matrix in Figure 4 has been indicated. The revised Fig. 4 is as follows: By marking epoxy and GO in the current TEM image does not help much as the image is of poor quality. It is highly recommended that authors should include the better quality TEM image to support their statements in the final manuscript. Some of the authors reply have not been included in the revised manuscript e.g., Response: As suggested. the nanosheets morphology of GO and EHBPE-GO were reanalyzed by TEM imaging, and the TEM images are shown as follows: The new images are almost the same as the previous images. We have made lots of efforts in analyzing the reason by consulting with the professor who specialized in TEM testing. The main reason may be the low glass transition (Tg) of EHBPE-GO. EHBPE-GO is viscous at room temperature, and the sample will collapse on the copper mesh, which results in the poor contrast. The other reason is the lower resolution of the instrument. 2. Is it epoxide-terminated (EHBPE)? Response: Yes, it is epoxide-terminated (EHBPE). - The reference material used in this comparative study was unmodified neat DGEBA resin. Include the above change. Response: As suggested. the above change has been added in the revised version, which are highlighted in page 3, line 16 and page 1, 18-19. Comment. 2: 1. Can the authors elaborate the advantages of hyperbranched polymers as filler? Response: The advantages of hyperbranched polymers as filler are as follows: the lower viscosity of polymers which can improve the processability; the more functional groups that can increase the crosslink density and reactivity. Include this in the manuscript Response: As suggested, the above change has been added in the revised version, which are highlighted in page 2, line 17-20. The added parts are as follows: “Therefore, when used as a filler, hyperbranched polymers show the lower viscosity of polymers which can improve the processability and the more functional groups that can increase the crosslink density and reactivity.”
